# Zirconia Hybrid Dental Implants Influence the Biological Properties of Neural Crest-Derived Mesenchymal Stromal Cells

**DOI:** 10.3390/nano14050392

**Published:** 2024-02-20

**Authors:** Nadia Tagliaferri, Alessandra Pisciotta, Giulia Orlandi, Giulia Bertani, Rosanna Di Tinco, Laura Bertoni, Paola Sena, Alice Lunghi, Michele Bianchi, Federica Veneri, Pierantonio Bellini, Jessika Bertacchini, Enrico Conserva, Ugo Consolo, Gianluca Carnevale

**Affiliations:** 1Department of Surgery, Medicine, Dentistry and Morphological Sciences with Interest in Transplant, Oncology and Regenerative Medicine, University of Modena and Reggio Emilia, 41124 Modena, Italy; nadia.tagliaferri@unimore.it (N.T.); giulia.orlandi@unimore.it (G.O.); giulia.bertani@unimore.it (G.B.); rosanna.ditinco@unimore.it (R.D.T.); laura.bertoni@unimore.it (L.B.); paola.sena@unimore.it (P.S.); federica.veneri@unimore.it (F.V.); pierantonio.bellini@unimore.it (P.B.); jessika.bertacchini@unimore.it (J.B.); enrico.conserva@unimore.it (E.C.); ugo.consolo@unimore.it (U.C.); gianluca.carnevale@unimore.it (G.C.); 2PhD Program in Clinical and Experimental Medicine, Department of Biomedical, Metabolic and Neural Sciences, University of Modena and Reggio Emilia, 41124 Modena, Italy; 3Center for Translational Neurophysiology of Speech and Communication, Fondazione Istituto Italiano di Tecnologia, 44121 Ferrara, Italy; alice.lunghi@iit.it; 4Section of Physiology, University of Ferrara, 44121 Ferrara, Italy; 5Department of Life Sciences, University of Modena and Reggio Emilia, 41125 Modena, Italy

**Keywords:** neural crest-derived MSCs, zirconia, dental pulp stem/stromal cells, dental implants

## Abstract

Dental implants are regularly employed in tooth replacement, the good clinical outcome of which is strictly correlated to the choice of an appropriate implant biomaterial. Titanium-based implants are considered the gold standard for rehabilitation of edentulous spaces. However, the insurgence of allergic reactions, cellular sensitization and low integration with dental and gingival tissues lead to poor osseointegration, affecting the implant stability in the bone and favoring infections and inflammatory processes in the peri-implant space. These failures pave the way to develop and improve new biocompatible implant materials. CERID dental implants are made of a titanium core embedded in a zirconium dioxide ceramic layer, ensuring absence of corrosion, a higher biological compatibility and a better bone deposition compared to titanium ones. We investigated hDPSCs’ biological behavior, i.e., cell adhesion, proliferation, morphology and osteogenic potential, when seeded on both CERID and titanium implants, before and after cleansing with two different procedures. SEM and AFM analysis of the surfaces showed that while CERID disks were not significantly affected by the cleansing system, titanium ones exhibited well-visible modifications after brush treatment, altering cell morphology. The proliferation rate of DPSCs was increased for titanium, while it remained unaltered for CERID. Both materials hold an intrinsic potential to promote osteogenic commitment of neuro-ectomesenchymal stromal cells. Interestingly, the CERID surface mitigated the immune response by inducing an upregulation of anti-inflammatory cytokine IL-10 on activated PBMCs when a pro-inflammatory microenvironment was established. Our in vitro results pave the way to further investigations aiming to corroborate the potential of CERID implants as suitable biomaterials for dental implant applications.

## 1. Introduction

Nowadays, the use of dental implants to replace missing teeth is a beneficial routine approach due to the reliable and long-lasting performance of common implant-supported prostheses. As a matter of fact, a huge amount of clinical research has been carried out to improve the efficacy of dental implants by identifying materials with optimal biological and mechanical properties and by tailoring their surface chemistry and topography to enable a better comprehension of the relationship between the characteristics of the implant surface and the alveolar bone and soft tissue biology. It is well established that both the osseointegration rate and the percentage of bone-to-implant contact (BIC) strongly rely on implant surface properties [1,2,3]. In particular, chemical composition, surface energy, wettability, roughness, zeta potential and topography at different length scales have been pointed out as crucial in favoring cell adhesion and survival and guiding osteogenic differentiation [3,4,5,6]. Traditionally, titanium has been considered the gold-standard material for oral and maxillofacial implants. It is a highly reactive metal that develops a very thin (about 2–10 nm) native passivating layer of titanium dioxide (TiO_2_) over its surface upon exposure to air or fluids [7], providing a crucial interface between the implant and the biological milieu. Such a native thin oxide layer confers suitable biocompatibility to the implant by simultaneously reducing material reactivity and partially preventing metallic corrosion, both clinically demonstrated to highly compromise survival rates and long-term stability of the implant [7]. Aside from the large success and diffusion of titanium implants, implant failure is still a challenge, since it accounts for about 5–11% within 10 to 15 years and entails the subsequent removal of the implants. Biomechanical and biological events, such as implant material degradation, allergic reactions and chronic peri-implant inflammation, have been identified as closely related to implant failure and loss. In general, peri-implantitis and titanium implant degradation can be considered as reciprocally correlated. As a matter of fact, dental implants are typically exposed to an environment characterized by the abundance of microorganisms, variable pH and chemical compounds, such as cetylpyridinium chloride, sodium fluoride and hydrogen peroxide [8]. Taken together, these factors can partially degrade the thin TiO_2_ layer, eventually leading to the establishment of corrosion pathways up to implant damage or even rupture of the material integrity. In this scenario, cells in direct contact with the exposed surface of the material are prompted to secrete inflammatory mediators, mainly neutrophils and macrophages [9]. Hence, alternative materials for dental implants are highly needed. Among possible candidates, zirconia has been revealed as a highly promising substitute for replacing titanium implants, mainly due to a higher resistance to pitting high mechanical loads and a low propensity to bacterial biofilm formation [10,11,12,13]. Interestingly, several zirconia-based coatings have been recently explored, showing suitable morphological, nanomechanical and tribological properties as well as an absence of cytotoxicity [14,15,16,17,18]. In addition, recent findings highlighted that zirconia proved to have a good biocompatibility due to the limited inflammatory response and bone loss when compared to titanium implants [19,20]. Human dental pulp stem cells (hDPSCs) represent a suitable cell source among the oral cavity stem cells for conducting in vitro studies on the interactions with dental implant materials due to their peculiar neural crest-origin-related properties, i.e., cell proliferation, differentiation potential and the expression of immunoregulatory molecules, which confer on them the capability to modulate the immune/inflammatory response and to preserve bone mass by modulating osteoclasts [21,22,23,24,25,26,27,28,29,30]. Based on this premise, the aim of the present study is to investigate the biological properties of hDPSCs when cultured on zirconia-coated titanium surfaces or on traditional titanium surfaces prepared according to different cleansing protocols by assessing cell adhesion, proliferation, osteogenic differentiation and inflammatory response.

## 2. Materials and Methods

### 2.1. Sample Preparation

A total of 60 titanium disks (Myplant, Parma, Italy) measuring 10 mm in diameter and 2 mm in thickness were utilized in this study. Specifically, two different titanium surfaces were examined: sandblasted/acid-etched (SLA) and zirconia-coated (CERID). Three experimental groups were defined: (1) control group, i.e., untreated surfaces (CTRL); (2) air-polishing-treated group (AP); (3) nickel–titanium-brush-treated group (BRUSH). Disks underwent cleansing treatment through two mechanical methods: (a) an air-polishing system (Combi-Touch, Mectron spa, Carasco, Genova, Italy) with 40 µm bicarbonate powder, and (b) nickel–titanium brushes (I.C.T. Brush micro, Hans Korea Co. Ltd., Gyeonggi-do, Korea/De Ore, Verona, Italy), as previously described [26]. Briefly, the I.C.T. nickel–titanium brushes were used at 400 rpm and 600 rpm, respectively, for two sequential rounds of 45” each, applying a 25 g pressure calibrated on an electronic scale with 100 N of torque. All the treatments were performed by the same operator under irrigation with buffer saline solution (0.9% NaCl). The “Combi-Touch” air-polishing system with sodium bicarbonate particles (ø 40 μm) was used for 30” with a 5 mm distance. The operating principle of this air-polishing system is based on spreading an accelerated flow of particles onto the titanium surface by means of compressed air. When the particles hit the surface, their kinetic energy is dissipated almost completely, resulting in a delicate but successfully cleansing effect. To complete the cleansing procedure, a water jet—arranged in a bell shape surrounding the main flow—is used to prevent the powder cloud from bouncing and being dismissed and, at the same time, to dissolve the powder by washing the surface [26].

### 2.2. Sample Characterization

Scanning electron microscopy analyses were carried out to preliminarily assess the surface morphology (EVO MA 10—Carl Zeiss, Oberkochen, Germany) [26]. The surface topography of pristine, air-polished and brushed titanium and CERID substrates was investigated by AFM using a Park System XE7 AFM system (Park System, Suwon, Republic of Korea) operating in noncontact mode in air and at room temperature. A premounted silicon cantilever with Al backside reflective coating, typical tip curvature radius ca. 7 nm, elastic constant ca. 26 Nm^−1^ and resonance frequency ca. 300 Hz (OMCL-AC160TS, Olympus Micro Cantilever, Tokyo, Japan) was used. For each sample, different sets of images of decreasing size (from 20 × 20 μm^2^ to 5 × 5 μm^2^) were acquired on different areas of the surface and analyzed by the Park System XEI software (XEI 2016, ver. 4.3; Park System, Suwon, Republic of Korea) to obtain the root mean square (RMS) roughness [31]. Finally, wettability of the different materials was evaluated by means of water contact-angle measurements. Images were acquired with a house-built contact-angle unit and analyzed with ImageJ (NIH, Bethesda, MD, United States) to extract contact-angle values. The final values were averaged over three different areas of each sample. Statistical analysis was performed with OriginPro 2020 (OriginLab Corporation, MA, USA), and values were considered statistically different for *p*-values < 0.01.

### 2.3. Immune Selection of Human DPSCs

This study was carried out using commercial human dental pulp stem cells (hDPSCs; CTIbiotech, Meyzieu-Lyon, France) cultured according to previously established protocols [32]. After cell expansion, a MACS^®^ separation kit (Miltenyi Biotec S.r.l, Bologna, Italy) was used to obtain a homogenous hDPSCs population through an immune selection against the surface antigens STRO-1 and c-Kit [32].

### 2.4. Cell Morphology and Proliferation

Undifferentiated STRO-1^+^/c-Kit^+^ hDPSCs were seeded on titanium and CERID disks (3 × 10^3^ cell/cm^2^) in 24-multiwell units and cultured under standard conditions for 24, 48, 72 and 96 h, then cell morphology was assessed by labelling cells with AlexaFluor546 Phalloidin (Abcam, Cambridge, UK). In order to analyze cell proliferation, 10 randomly selected fields with a 4.05 × 10^5^ µm^2^ surface area per disk (n = 3 disks) for each experimental group were chosen, and nuclei, previously stained with 1 µg/mL 40,6-diamidino-2-phenylindole (DAPI), were counted by a blind operator and acquired by means of a Nikon A1 confocal fluorescence microscope (Nikon, Tokyo, Japan), as formerly described by Di Tinco et al. [26].

### 2.5. Evaluation of Neural Crest Markers in hDPSCs Cultured on Titanium Disks

In order to evaluate the biological properties of hDPSCs seeded on titanium and titanium–zirconia (CERID) disks before and after treatment, stemness markers were analyzed. Particularly, after 72 h of culture, cells were fixed and permeabilized as described above. After blocking with 3% bovine serum albumin (BSA) in PBS, cells were immunolabeled with mouse anti-nestin (Merck Millipore, Burlington, MA, United States) and rabbit anti-SOX10 (Abcam, Cambridge, UK) primary antibodies and subsequently targeted by AlexaFluor-conjugated secondary antibodies (Thermo Fisher Scientific, Waltham, MS, USA) [26]. A Nikon A1 confocal laser scanning microscope was used to perform confocal imaging. Confocal serial sections were processed with Fiji ImageJ software (Fiji for Mac OS X; NIH, Bethesda, MD, USA) to obtain 3-dimensional projections, and image rendering was conducted with Adobe Photoshop software (Adobe Photoshop 2022, ver. 23.5.2) [26].

### 2.6. Osteogenic Induction

In order to evaluate the ability of SLA titanium and CERID surfaces to affect osteogenic differentiation, hDPSCs were seeded onto disks (3 × 10^3^ cells/cm^2^) and cultured under standard conditions for 72 h. Subsequently, standard culture medium was replaced with osteogenic medium (α-MEM, 10% FBS, 2 mM L-glutamine, 100 U/mL penicillin, 100 mg/mL di streptomycin, 100 nM dexamethasone, 10 mM di β-glycerophosphate, 100µM ascorbic acid, all from Sigma-Aldrich, St. Louis, MO, USA). After 3 weeks of induction, the expression of osteogenic markers, RUNX2, osteopontin (OPN) and osteocalcin (OCN), was assayed through immunofluorescence analyses. Rabbit anti-RUNX2, mouse anti-OPN and mouse anti-OCN (Abcam, Cambridge, UK) primary antibodies were incubated at a 1:50 dilution and revealed with anti-rabbit AlexaFluor546 and anti-mouse Alexa fluor488 secondary antibodies (1:200; Thermo Fisher Scientific). Nuclei were counterstained with DAPI [26].

### 2.7. PBMCs Isolation and Culture in Presence of Titanium Surfaces

Peripheral blood mononuclear cells (PBMCs) were isolated from the fresh whole blood of healthy adult volunteers (n = 5) by using Histopaque^®^ (Sigma Aldrich, St. Louis, MO, USA) according to the manufacturer’s instructions and preactivated by adding anti-CD3 and the costimulatory anti-CD28 monoclonal antibodies (BD Biosciences, Franklin Lakes, NJ, USA) to the culture medium [32]. PBMCs, resting or preactivated, according to each experimental evaluation, were then seeded and cultured in 6-well plates (3.5 × 10^6^ cell/well) in the presence of both SLA titanium and CERID disks in order to investigate the biological effects of both implants on PBMCs. After 72 h of culture, the floating PBMCs were collected together with the supernatant and processed according to the subsequent experimental procedures. Resting PBMCs and preactivated PBMCs cultured alone were used as controls.

### 2.8. RNA Purification and Quantitative Real-Time PCR

The biological response of human PBMCs was evaluated by analyzing IL-2, IL-6, IL-10, IFNγ and TNFα through RT-PCR. Cells were homogenized and total RNA was extracted and purified using TRIzol (Thermo Fisher Scientific, Waltham, MA, USA), then RNA integrity and quantification were assessed through a spectrophotometric method by means of a NanoDrop 2000 device (Thermo Fisher Scientific, Waltham, MA, USA). Total RNA (1 μg) was reverse-transcribed to cDNA using a QuantiTect Reverse Transcription Kit (Qiagen, Hilden, Germany) according to the manufacturer’s instructions [29]. Quantitative real-time PCRs were performed on a QuantStudio™ 3 Real-Time PCR System (Applied Biosystems, Thermo Fisher Scientific, Waltham, MA, USA) using the PowerTrack SYBR Green Master Mix (Thermo Fisher Scientific, Waltham, MA, USA) with different oligonucleotides (Sigma Aldrich, St. Louis, MO, United States; Table 1). The relative gene expression quantification was performed using the comparative threshold (Ct) method (ΔΔCt), where relative gene expression level equals 2^−ΔΔCt^. The obtained fold changes in gene expression were normalized to the housekeeping gene RPLP0 [32].

### 2.9. Statistical Analysis

Cell proliferation was analyzed by one-way ANOVA followed by Newman–Keuls post hoc test. The real-time PCR data were analyzed by one-way ANOVA followed by Dunnett post hoc test (GraphPad Prism Software version 5 Inc., San Diego, CA, USA). In all cases, the values were expressed as the mean ± standard deviation from three independent experiments. For all tested groups, the statistical significance was set up at *p* < 0.05.

## 3. Results

### 3.1. Surface Characterization

SEM and AFM images acquired from the different experimental groups are shown in Figure 1 and Figure 2, respectively. The bare titanium surface appears more homogeneous than the bare CERID surface, as the latter is characterized by the presence of microsized grains with nonuniform dimensions (Figure 1D and Figure 2B).

The surface morphology of bare titanium, instead, is characterized by the presence of a micrometric ripple with alveolus-like morphology (Figure 1A and Figure 2A). Air polishing seems not to affect sample morphology to any great extent, either in the case of CERID (Figure 1E and Figure 2D) or titanium (Figure 1B and Figure 2C). Interestingly, both materials appeared to be altered by the brushing step, although to a different extent. Indeed, while polishing scratches and linear marks are well visible on the titanium samples, leading to the loss of the pitted structure and to an overall increase in surface roughness (Figure 1C and Figure 2E), more subtle effects can be found on CERID, such as the formation of quite flat, large grains (Figure 1F and Figure 2F).

Analysis of AFM topographical images allows us to gain quantitative information about such surface modification, for instance the evolution of surface roughness (expressed here as root mean square roughness, RMS) at different length scales (Figure 2G). As can be observed, the effect of brushing on the titanium surface is reflected in a net RMS increase, more pronounced where large scan sizes are considered as the latter better allow us to intercept major polishing marks. On the contrary, RSM was found to decrease on the CERID surface after the polishing treatments, likely due to the onset of large, smooth splats. Contact-angle measurements carried out on titanium and CERID samples (Figure 2H) pointed out a similar wettability for the two materials, both being hydrophilic surfaces. Hydrophobicity was found to decrease upon the polishing treatments independent from the material considered. Taken together, these data suggest that cleaning the surface of both CERID and titanium substrates with air polishing leads to negligible changes of the surface microstructure (as proven by similar values of RMS and water contact angle). Brush polishing is more aggressive on the surface, affecting in particular surface roughness, albeit showing an opposite effect on the two materials. The formation of smooth splats on the CERID surface points to a more malleable behavior of this substrate compared to titanium, most likely due to the different chemical natures of titanium and CERID. Overall, CERID surface features are less affected by the polishing method compared to a standard titanium surface.

### 3.2. Cell Morphology and Cell Proliferation of hDPSCs

In order to evaluate the morphology, adhesion and proliferation of hDPSCs cultured on disks, cells were seeded on titanium and CERID surfaces after cleansing with AP and BRUSH. Experiments were conducted at different times (24 h, 48 h, 72 h, 96 h). As shown in Figure 3, the AP treatment did not affect the cell morphology of hDPSCs grown on titanium disks, which displayed a fibroblast-like shape, whereas the treatment with the brush cleansing system resulted in a more aligned cell growth fashion along the grooves when compared to the control group (Figure 4), in accordance with data obtained by SEM and AFM analyses.

With regard to CERID surfaces, neither AP nor BRUSH treatments appeared to alter cell morphology, which showed a spindle-like appearance in all the experimental groups. The proliferation rate of hDPSCs cultured on titanium and CERID, either cleansed or not, is shown in Figure 5. As can be observed, both cleansing treatments sharply increased the proliferation rate of hDPSCs grown on titanium disks after 96 h of culture (*** *p* < 0.001 TIT AP vs. TIT CTRL; °°° *p* < 0.001 TIT BRUSH vs. TIT CTRL). In contrast, the cleansing systems applied to CERID surfaces did not affect the cell proliferation rate at any evaluated experimental time (Figure 5B).

### 3.3. Evaluation of Stemness and Neural Crest Markers

The expression of the neural crest stem cell-related markers SOX10 and nestin was investigated in hDPSCs cultured on titanium and CERID after cleansing treatments in order to assess the stemness maintenance. As shown in Figure 6, the cleansing treatments did not affect the expression or the localization of the investigated markers in any of the experimental groups, suggesting that both surfaces, cleansed or not, proved suitable in maintaining the neural crest-related stemness phenotype of hDPSCs.

### 3.4. Osteogenic Differentiation

The potential effects of both titanium and CERID disks on the capability of hDPSCs to commit towards the osteogenic lineage were investigated by culturing cells with appropriate inductive stimuli for 3 weeks. Then, immunofluorescence analysis was performed to analyze the expression of the bone extracellular matrix-associated markers OCN, OPN and RUNX-2. As reported in Figure 7, cells cultured on disks without inductive stimuli (CTRL) showed a mild expression of OCN, OPN and RUNX-2 when cultured on both surfaces, suggesting that titanium and CERID possess an intrinsic potential to promote the osteogenic commitment of hDPSCs. When cells were seeded on disks in the presence of the differentiation medium, the expression of the bone extracellular matrix-associated markers was even stronger, with OCN, OPN and RUNX-2 being expressed with higher intensity in hDPSCs cultured on CERID rather than the counterparts seeded on titanium (Figure 7A,B). These data suggest the suitability of the CERID surface in supporting the osteogenic differentiation processes.

### 3.5. Cytokine Production in an In Vitro Model of Inflammation

The experimental setup was carried out to study the pro-inflammatory properties of both materials by culturing them with resting PBMCs (rPBMCs). At the same time, we performed a coculture with preactivated PBMCs (aPBMCs) to mimic an established inflammatory microenvironment resembling the pathological conditions. As shown in Figure 8A, rPBMCs cultured with both materials did not show any significant differences in the expression of pro-inflammatory cytokines, suggesting that biomaterials per se did not exert pro-inflammatory properties.

Instead, when aPBMCs were cultured on CERID, a statistically significant upregulation of IL-6 was observed in parallel with a significant downregulation of TNF-α when compared to aPBMCs alone (** *p* < 0.01). At the same time, a statistically significant upregulation of IL-10 was observed in aPBMCs when cultured with CERID in comparison to aPBMCs cultured on titanium (* *p* < 0.05). These data confirm that CERID did not exacerbate a pro-inflammatory status but rather modulated the immune cells’ involvement by favoring a regulatory immune milieu.

## 4. Discussion

Titanium is considered so far to be one of the best materials in implant applications since it possesses remarkable properties, such as high fatigue and corrosion resistance in biological fluids, thanks to its thin TiO_2_ layer [33]. It is well established that several surface parameters, including wettability, roughness, local topography, zeta potential, surface energy and chemistry, are pivotal in positively affecting the biocompatibility, osseointegration and, thus, long-term durability and stability of dental implants by promoting cell adhesion and proliferation and, ultimately, high bone-to-implant contact [3,4,5,6]. In general, higher roughness is associated with increased cell colonization of the implant surface and accelerated osseointegration [34,35]. However, it should be pointed out that highly rough titanium surfaces can also promote bacterial biofilm accumulation, providing the basis for the onset of peri-implantitis [13,36,37] and eventually leading to corrosion of the TiO_2_ layer and damage/failure of the implant itself. Such degradation processes are thought to involve the release of titanium particles that, in turn, trigger the recruitment of immune cells secreting inflammatory factors, leading to bone resorption (osteolysis) and, eventually, implant loosening. Alternative materials to titanium have been investigated to overcome this issue. Among them, zirconia has been extensively used in medical applications such as implant abutments and joint replacement appliances, demonstrating good biocompatibility and low affinity for plaque besides suitable mechanical and chemical properties, making zirconia an ideal candidate for dental implants [38,39]. To this end, our study aimed to evaluate and compare the effects of titanium and hybrid titanium–zirconia surfaces on the biological properties of hDPSCs. In order to mimic cleansing effects on the implant surface, both air-polishing and brush treatments were carried out. SEM analysis indicated that the air-polishing treatment did not significantly alter the surface morphology, whereas titanium disks were mostly affected by the brushing treatment compared to CERID. AFM analysis confirmed this evidence, pointing out an increase in roughness on titanium disks undergoing brush polishing, which is in accordance with previous findings [40]. On the contrary, surface roughness was found to slightly decrease for CERID due to the appearance of smooth grains after the brush step. Regarding surface wettability, it is well-known that in general, moderate or highly hydrophilic surfaces (50°–70°) are suitable for cell adhesion compared to hydrophobic (>90°) or extremely hydrophilic (<10°) surfaces [41,42,43]. Notably, all measured contact-angle values fall well within the range of wettability considered optimum for cell adhesion and proliferation, especially after the polishing steps. This can be ascribed to the generation of large, almost flat areas both in the titanium (between one scratch mark and the closest one) and in the CERID (due to the generation of smooth splats). It is worth stressing that air polishing and brushing were selected as cleansing methods because they both rely on physical treatments rather than on chemicals, thus reducing the probability of altering the chemical properties of titanium and CERID surfaces. However, other effects cannot be completely excluded at this stage, such as the introduction of hydrophilic groups (-OH) upon the polishing steps, that could further contribute to the overall decrease in the contact-angle value. The effects of brush treatment also reflected a shift in cell morphology and alignment on the titanium surface, while hDPSCs adhered and grew on CERID surfaces, maintaining their typical spindle-like shape. In parallel, the cell proliferation rate of hDPSCs sharply increased with culture time when cells were grown on titanium surfaces, after AP and BRUSH cleansing, achieving a peak after 96 h of culture. Conversely, the cleansing treatments did not induce any significant differences in the cell proliferation trend of hDPSCs cultured on CERID. Furthermore, when analyzing the expression of the neural crest-related markers nestin and SOX10, titanium and CERID showed comparable results, demonstrating that both surfaces were able to maintain the stemness properties of hDPSCs, even after different cleansing procedures. Taken together, these data highlight that the evaluated materials did not alter the biological properties and neural crest-related features of hDPSCs, thus confirming their good biocompatibility in accordance with previous findings [44,45,46]. Data regarding the osteogenic differentiation of hDPSCs cultured on titanium and CERID underlined that both materials hold an intrinsic potential to promote osteogenic commitment of neuro-ectomesenchymal stromal cells, as shown by a mild expression of early and late bone-tissue-related markers, yet in cells grown without the addition of osteogenic stimuli. Particularly, in hDPSCs differentiated on the CERID surface, the immunolabeling against proteins of the extracellular mineralized matrix showed a remarkable intensity, underpinning the suitability of this surface in supporting the progression of osteogenic commitment. Dental implant features such as topography, wettability and porosity are also key factors in the control of the immune response. The contact between exogenous materials and immune system cells initiate the production of a cytokine cascade, which could lead either to a prohealing response or towards the development of chronic inflammation with resulting bone loss and implant failure [47,48]. It is noteworthy that the analyzed materials did not induce the release of pro-inflammatory cytokines by resting PBMCs, confirming that titanium and CERID do not directly activate the immune response. Further evidence emerged when an established pro-inflammatory microenvironment was mimicked by culturing activated PBMCs with either titanium or CERID surfaces. As a matter of fact, our data pointed out that aPBMCs cultured with both implant surfaces showed a similar response by upregulating IL-6 and, at the same time, downregulating TNFα, suggesting that the increased expression of the pleiotropic cytokine IL-6 reflected a survival-promoting role in aPBMCs themselves. Interestingly, only the CERID surface was able to induce a significant upregulation of the anti-inflammatory cytokine IL-10 in aPBMCs, hinting that this implant surface not only mitigated a preexisting pro-inflammatory microenvironment but also modulated the immune cells by promoting a regulatory immune scenario. These findings might prompt the suitability of CERID as a worthy biomaterial for dental implant applications.

## 5. Conclusions

The present study demonstrated that titanium and hybrid titanium–zirconia, i.e., CERID, are suitable materials for the adhesion, proliferation and osteogenic differentiation of hDPSCs under the appropriate stimuli besides maintaining their neural crest-related stemness phenotype under standard culture conditions. After cleansing procedures with brushing or air-polishing systems were performed on titanium and CERID surfaces, no alterations were recorded in the biological features of hDPSCs, thus reflecting no significant modification of both surfaces in terms of surface morphology and nanotopography. Only a slight increased surface roughness was observed on titanium disks after brush treatment. Our data showing the capability of CERID to modulate a pro-inflammatory microenvironment are in accordance with previous findings in the literature, supporting the hypothesis that zirconia might hold a protective effect against inflammation and thus represent a suitable alternative to titanium implants in patients more prone to peri-implantitis. Further investigations are needed to deeply corroborate the potential of zirconia surfaces, with particular regard to long-term follow-up studies.

## Figures and Tables

**Figure 1 nanomaterials-14-00392-f001:**
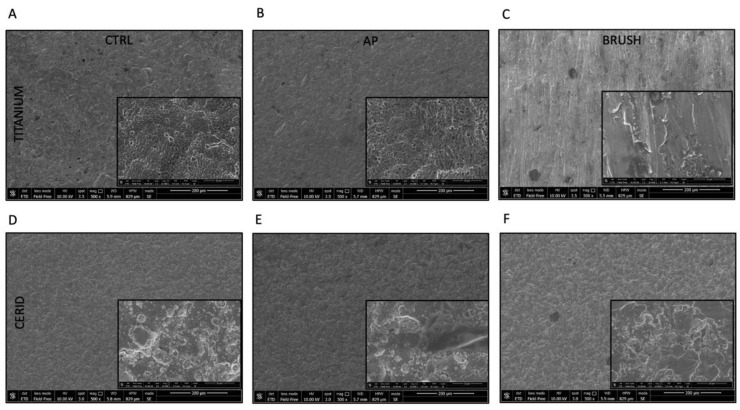
Titanium and CERID surface nanotopography. SEM analysis was carried out on both surfaces from three experimental groups (CTRL, AP, BRUSH) in order to evaluate the surface topography; inserts in rectangles show 10,000× magnifications. CTRL = control, AP = air-polishing treatment, BRUSH = nickel–titanium brush treatment. White arrows highlight the presence of grooves on the titanium surface after brush treatment. Scale bars: 200 µm (images) and 10 µm (magnification inserts). (**A**) Titanium ctrl; (**B**) Titanium AP; (**C**) Titanium Brush; (**D**) CERID ctrl; (**E**) CERID AP; (**F**) CERID Brush.

**Figure 2 nanomaterials-14-00392-f002:**
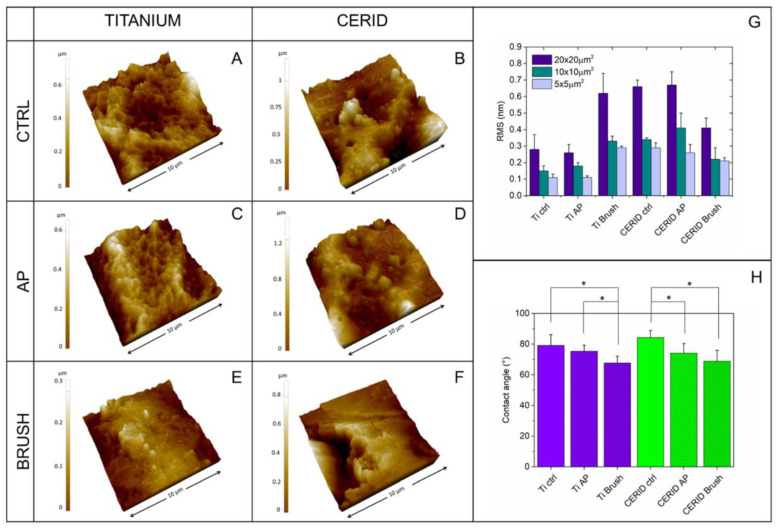
Results of AFM and wettability analyses. Three-dimensional topography (10 × 10 µm^2^) of titanium (**A**,**C**,**E**) and ceramic-coated (**B**,**D**,**F**) samples and root mean square roughness (RMS) values extracted from different scan sizes (**G**). Water contact-angle measurements are reported in panel (**H**). * *p* < 0.01 Ti ctrl vs. Ti Brush, Ti AP vs. Ti brush, * *p* < 0.01 CERID ctrl vs. CERID AP, CERID ctrl vs. CERID Brush.

**Figure 3 nanomaterials-14-00392-f003:**
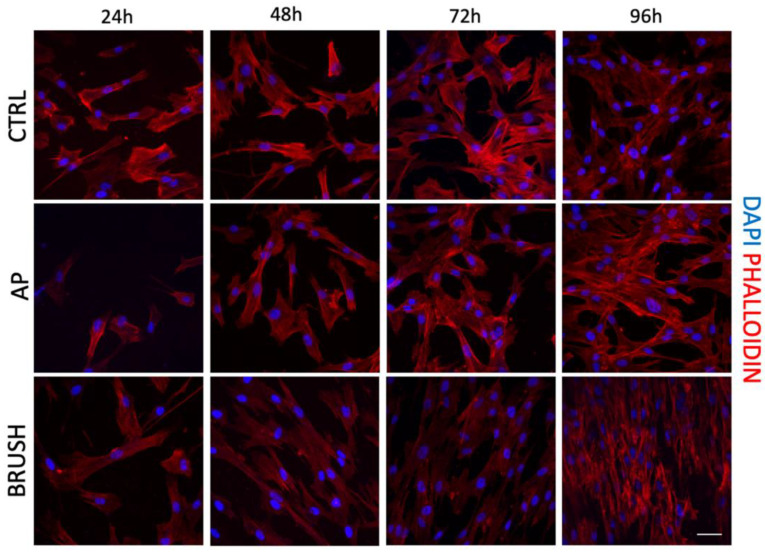
Immunofluorescence analysis of phalloidin (PHA)-stained (red) hDPSCs cultured at different experimental time points (24 h, 48 h, 72 h, 96 h) on titanium disks formerly treated with air-polishing (AP) or brush (BRUSH) cleansing systems. Cells seeded on uncleansed titanium disks were used as control. Nuclei were counterstained with DAPI. Scale bar: 20 μm.

**Figure 4 nanomaterials-14-00392-f004:**
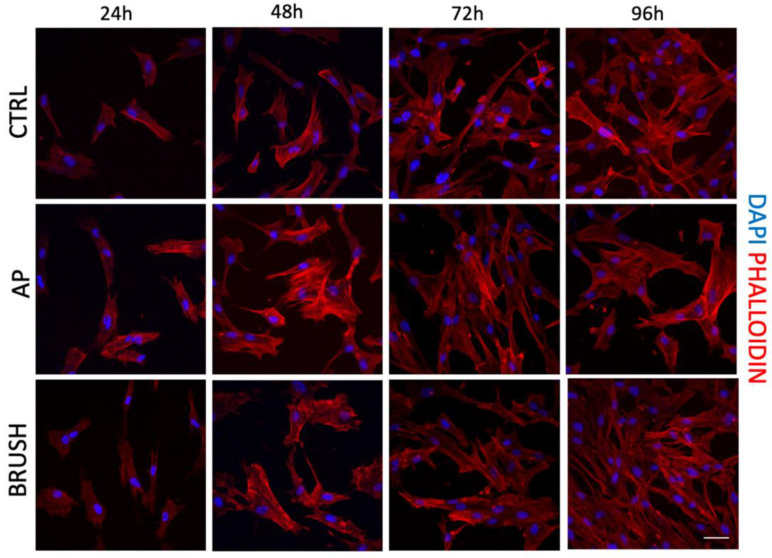
Immunofluorescence analysis of phalloidin (PHA)-stained (red) hDPSCs at different experimental time points (24 h, 48 h, 72 h, 96 h) on CERID disks previously treated with air-polishing (AP) or brush (BRUSH) cleansing systems. Control group (CTRL) consists of hDPSCs seeded on untreated CERID surfaces. Nuclei were counterstained with DAPI. Scale bar: 20 μm.

**Figure 5 nanomaterials-14-00392-f005:**
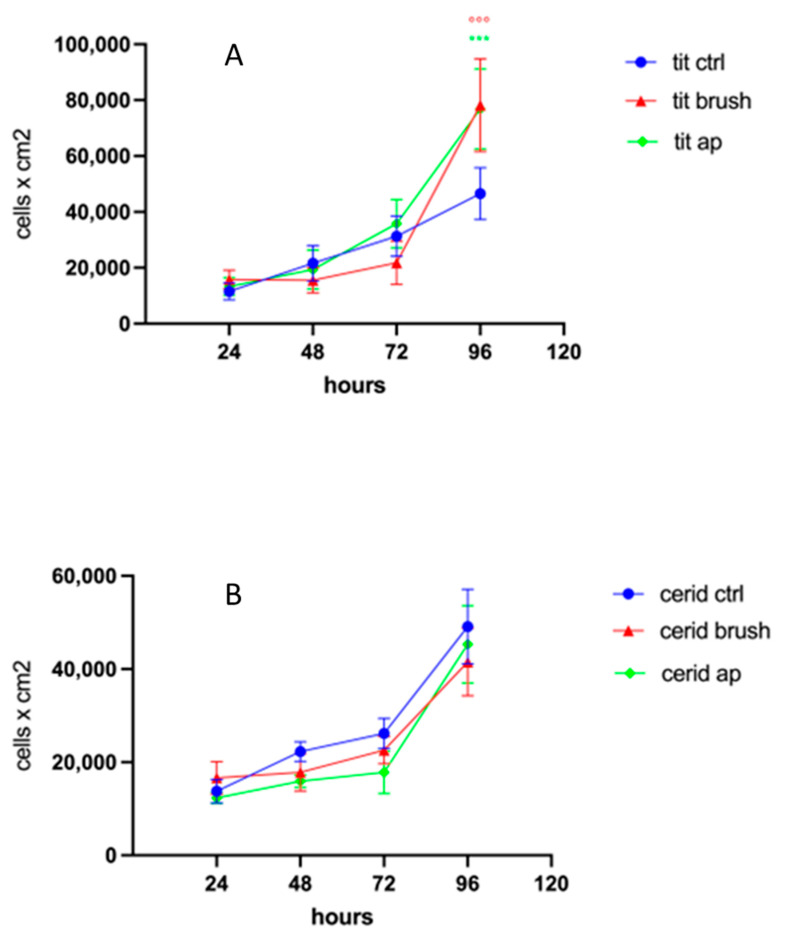
Line graphics representing proliferation rate of hDPSCs cultured at 24 h, 48 h, 72 h and 96 h on the three experimental groups examined (CTRL, brush and air polishing). (**A**) Cell proliferation rate of cells seeded on titanium disks untreated (CTRL) or cleansed (AP and BRUSH). (**B**) Cell proliferation rate of cells seeded on CERID disks untreated (CTRL) or cleansed (AP and BRUSH). Data are reported as mean ± SD, and statistical analysis was carried out by one-way ANOVA followed by Newman–Keuls post hoc test. *** *p* < 0.001 TIT AP vs. TIT CTRL; °°° *p* < 0.001 TIT BRUSH vs. TIT CTRL.

**Figure 6 nanomaterials-14-00392-f006:**
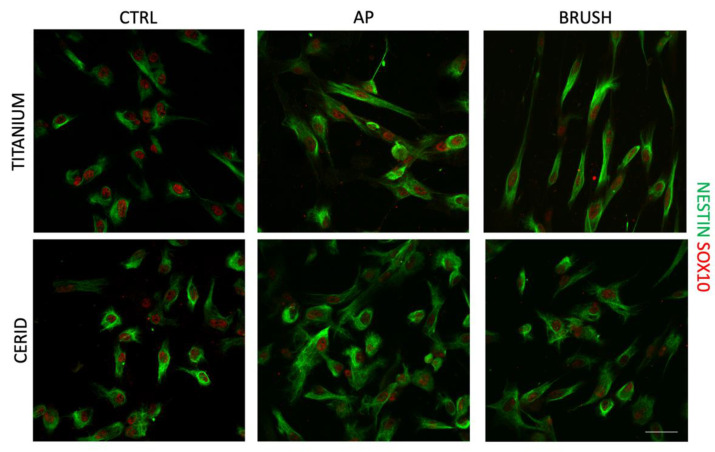
Immunofluorescence analysis of neural crest markers nestin (green) and SOX-10 (red) after 72 h of culture on hDPSCs seeded on titanium or CERID disks previously cleansed with air polishing (AP) and brush. Untreated titanium or CERID surfaces were used as control (CTRL). Scale bar: 20 μm.

**Figure 7 nanomaterials-14-00392-f007:**
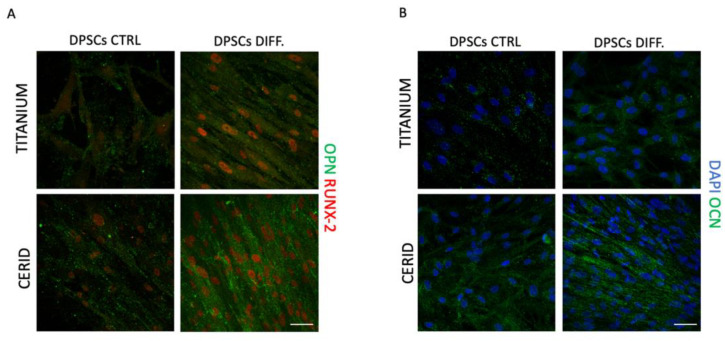
Evaluation of osteogenic differentiation of hDPSCs on titanium and CERID disks. Immunofluorescence analysis of osteopontin and osteocalcin (OPN, OCN; green) and RUNX-2 (red) in hDPSCs after 21 days of osteogenic induction on titanium (**A**) and CERID (**B**) disks. hDPSCs cultured without osteo-inductive stimuli were used as control (CTRL). Scale bar: 20 μm.

**Figure 8 nanomaterials-14-00392-f008:**
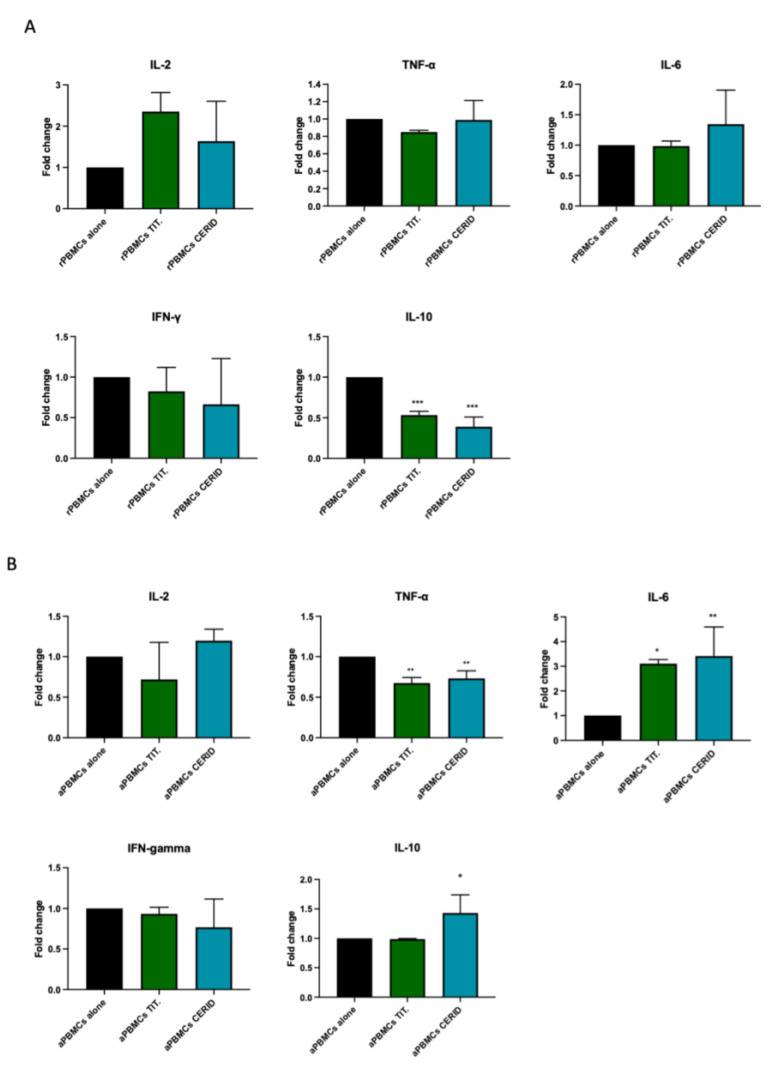
Evaluation of immune cells’ response to titanium and CERID surfaces. Real-time PCR analyses of IL-2, TNF-α, IL-6, IFN-γ and IL-10 in resting PBMCs (**A**) and in activated PBMCs (**B**). Histograms showing fold change in mRNA expression levels of each cytokine. Data are represented as mean ± SD and statistical analysis was carried out by one-way ANOVA followed by Dunnett post hoc test (*** *p* < 0.001 vs. rPBMCs alone; * *p* < 0.05, ** *p* < 0.01, *** *p* < 0.001 vs. aPBMCs alone).

**Table 1 nanomaterials-14-00392-t001:** List of the oligonucleotides used in real-time PCR analyses.

Target Gene	Forward Sequence	Reverse Sequence
hRPLP0	TACACCTTCCCACTTGCTGA	CCATATCCTCGTCCGACTCC
hIL-2	AAAGAAAACACAGCTACAACTGG	GAAGATGTTTCAGTTCTGTGGC
hIFNγ	GCATCGTTTTGGGTTCTCTTG	AGTTCCATTATCCGCTACATCTG
hTNFα	ACTTTGGAGTGATCGGCC	GCTTGAGGGTTTGCTACAAC
hIL-6	CCACTCACCTCTTCAGAACG	CATCTTTGGAAGGTTCAGGTTG
hIL-10	CAGAGTGAAGACTTTCTTTCAAATG	CCTTTAACAACAAGTTGTCCAGC

## Data Availability

The data that supported the findings of this study are available from the corresponding authors upon reasonable request.

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
