# Peer review of "Zirconia Hybrid Dental Implants Influence the Biological Properties of Neural Crest-Derived Mesenchymal Stromal Cells"

_nanomaterials, 2024, doi:10.3390/nano14050392_

Round 1
Reviewer 1 Report
Comments and Suggestions for Authors
This study is overall well-conducted but suffers from majors flaws.
1. The most important one is the rationale behind it. Considered implants are used when there is no longer safe pulp and therefore no DPSCs. Moreover DPSCs, to my knowledge, have not been shown to contribute to implant osseointegration (although they have an osteogenic potential). I also noted that refs 21 and 22 do not refer to DPSCs. Thus, while the work is interesting per se, the authors must revise the introduction section and provide a sound scientific rationale for conducting it
2. There is too little references related to DPSCs and their osteogenic potential. Even the seminal work by Gronthos et al. is missing
3. Please improve the contrast of images Fig 1. Also insert lettered labels that are currently missing
4. Fluorescent images of immunostained samples (Fig 6,7) are of poor quality, with numerous very bright spots. Please prepare cleaner samples
5. Fold changes in PCR data are relatively small (2>x>0.5) in most cases. The authors should therefore be much more cautious about their conclusions.
6. A discussion of the difference of behavior between the two surfaces under the various treatments is lacking, based on the chemical and physical properties of the two materials (TiO2 vs TiO2-ZrO2)
7. related to point 1, what kind of dental application would benefit from the results of the present work ?
Comments on the Quality of English LanguageThe english syntax can easily be improved.
Author Response
Comments and Suggestions for Authors
This study is overall well-conducted but suffers from majors flaws.
1. The most important one is the rationale behind it. Considered implants are used when there is no longer safe pulp and therefore no DPSCs. Moreover DPSCs, to my knowledge, have not been shown to contribute to implant osseointegration (although they have an osteogenic potential). I also noted that refs 21 and 22 do not refer to DPSCs. Thus, while the work is interesting per se, the authors must revise the introduction section and provide a sound scientific rationale for conducting it
Re: We thank the reviewer for the observations. The choice of using DPSCs in the present study relies on their biological features which strongly relate to the embryological origin from neural crest, which gives rise to hard and soft tissues constituting the maxillofacial complex. To this regard, previous literature demonstrates that DPSCs may provide a suitable stem cell source for conducting in vitro studies aimed at evaluating the response of neural crest derived MSCs to different investigated implant surfaces, based on their cell proliferation, osteogenic differentiation potential and immunomodulatory abilities towards an inflammatory microenvironment (doi: 10.1177/0022034512458690; doi: 10.3389/fphys.2018.00547; doi: 10.1186/s13287-021-02664-4). Our study aimed to compare the effects of titanium and zirconia surfaces, which have been used as implant materials in dentistry so far, using a peculiar source of mesenchymal stromal cells. This is the scientific rationale behind the choice of using DPSCs, although the clinical application of autologous DPSCs in combination with titanium or zirconia implant surfaces in regenerative dentistry does not fall within the scopes of our study nor was mentioned anywhere. Introduction section has been edited and proper references on DPSCs have been added, according to the reviewer's suggestion.
2. There is too little references related to DPSCs and their osteogenic potential. Even the seminal work by Gronthos et al. is missing.
Re: we thank the reviewer for the comment. We have added further references on DPSCs biological features in the introduction section.
3. Please improve the contrast of images Fig 1. Also insert lettered labels that are currently missing.
Re: the corrections requested by the reviewer have been made accordingly and a new figure 1 has been included along the text. It is likely that the resolution of the manuscript provided by the journal for the review process is lower than the one of the original format.
4. Fluorescent images of immunostained samples (Fig 6,7) are of poor quality, with numerous very bright spots. Please prepare cleaner samples.
Re: Thank you for your observations. Image of titanium ctrl with bright spots has been replaced with a new one in Figure 6. With regard to Figure 7, the immunolabeling against the evaluated osteogenic markers is well established in literature as it is shown in our data. The poor quality might be attributed to the lower resolution of the manuscript provided by the journal for the review process.
5. Fold changes in PCR data are relatively small (2>x>0.5) in most cases. The authors should therefore be much more cautious about their conclusions.
Re: we thank the reviewer for the comments. We agree with the reviewer on the fact that the fold change differences are small, however the statistical significance is related to the sample size used in the study. Of course these significant values do not strictly reflect the biological response.
6. A discussion of the difference of behavior between the two surfaces under the various treatments is lacking, based on the chemical and physical properties of the two materials (TiO2 vs TiO2-ZrO2)
Re: We thank the reviewer for the observation. The focus of the study is the effect of the cleansing methods on cell behavior rather than the effect on the substrates’ surface properties, either chemical or physical. Studying the effect of air polishing and brushing on titanium and CERID surfaces, however, was necessary to understand if upon cleansing substrates were still providing a permissive environment for cell adhesion or not, and if potential modification of the surface were correlated to changes in cell behavior. We would like to stress that an exhaustive characterization of surface modifications upon cleansing treatments and an accurate study about the extent to which the respective chemical nature impacts on the latter is beyond the scope of this study. However, we further expanded the comment on how air polishing and brushing affect surface roughness and wettability (see result section 3.1). Finally, we acknowledge that not only physical parameters but also chemical ones influence cell adhesion. Nonetheless, as air polishing and brushing only rely on physical agents (air laminar flux and mechanical forces) to clean the surface, we decided to solely focus on their effect on substrates’ physical properties rather than on chemical ones. In any case, we added a sentence to clarify this choice in the discussion paragraph.
7. related to point 1, what kind of dental application would benefit from the results of the present work ?
Re: we thank the reviewer for the comment. As indicated in the aim of our study, our findings may provide insights on the effects of dental surfaces alternative to titanium, such as CERID, on the progenitor mesenchymal stromal cells involved in the osseointegration processes occurring after implant dentistry procedures. In particular, the good biocompatibility demonstrated by CERID in maintaining the biological properties of cell proliferation and osteogenic differentiation of the studied neural crest derived stem cell population, support the choice of valuable alternative materials, such as zirconia surfaces, for regenerative dentistry. In any case, the materials used in the present study are implant materials already available on the market.
Comments on the Quality of English Language
The english syntax can easily be improved.
Re: we thank the reviewer for the recommendation and the english syntax has been reviewed accordingly.
Reviewer 2 Report
Comments and Suggestions for Authors
The paper presents a biological response of two materials used in dentistry subjected to different cleaning procedures. There are a few points to be addressed:
Please add the cleaning procedure shortly, even if you gave the references, it is better to include a short paragraph
Figures must be improved, scales are not visible, values are not readable and not given in the text
Given the type of surfaces, roughness profilometry on larger area should indicate the correct results
Contact angle should be coupled with surface energy measurements as well
Results are not discussed in comparison to other reported studies in the literature
Discussions should be separated and Conclusions should be added.
Discussions on the observed phenomena should be included, why surface chemistry did not influence or in which way, and how the modified CERID was better.
Author Response
Comments and Suggestions for Authors
The paper presents a biological response of two materials used in dentistry subjected to different cleaning procedures. There are a few points to be addressed:
Please add the cleaning procedure shortly, even if you gave the references, it is better to include a short paragraph
Re: we thank the reviewer for the comment, however journal instructions recommend a brief description of the experimental procedures. In any case, the procedure has been further detailed, accordingly.
Figures must be improved, scales are not visible, values are not readable and not given in the text
Re: we thank you for the observation, however the poor resolution might be due to the reformatting of the whole manuscript for review process by the journal. In any case, figures at high resolution have been provided as well.
Given the type of surfaces, roughness profilometry on larger area should indicate the correct results
Re: We thank the reviewer for the comment. Changes in surface roughness were evaluated to understand if the cleansing treatments create an environment that discourages cell adhesion. Therefore, as cells sense the surrounding environment rather than the entire surface of the sample, we investigated changes in local roughness. Also, we would like to remark that a characterization of surface roughness at different length scales is provided.
Contact angle should be coupled with surface energy measurements as well
Re: We thank the reviewer for the comment. We characterized the water contact angle of titanium and CERID surfaces to understand if cleaning protocols affected surface wettability and, consequently, their capability to support cell adhesion and colonization. We acknowledge that surface energy is among the parameters capable of influencing cell adhesion; however, providing an exhaustive characterization on physical and chemical modification of substrates surface upon cleansing treatments was beyond the scope of the study. For this reason, we carried out the study selecting only two figures of merits, namely wettability and surface roughness, though we are aware that also other parameters might influence cell adhesion.
Results are not discussed in comparison to other reported studies in the literature. Discussions should be separated and Conclusions should be added.
Re: we thank the reviewer, we have revised our discussion and added a conclusions section as recommended.
Discussions on the observed phenomena should be included, why surface chemistry did not influence or in which way, and how the modified CERID was better.
Re: we thank the reviewer for the comment. We are aware that surface chemistry plays a role in influencing cell adhesion and growth; indeed, the behavior of DPCs cultured on untreated titanium and on untreated CERID was investigated at different culture times (24, 72 and 96 hours). Results showed no significant differences in cells cultured on titanium or CERID, being this finding in good agreement with literature. For this reason, we are confident in considering negligible the effect of the different surface chemistry. Furthermore, as titanium and CERID substrates were solely treated with physical polishing methods, we assumed that the surface chemistry was not altered by the cleaning procedures. However, to better clarify this point, a sentence was added in the discussion paragraph.
Round 2
Reviewer 1 Report
Comments and Suggestions for Authors
Although I am still not convinced by the justification of this work nor by the lack of more detailed discussion of the properties of the tested materials, I acknowledge the overall seriousness of the work.
My only but big concern remains the literature citation. While I had previously mentioned the lack of references in the field of DPSCs, the authors have mainly added their own papers. This must be changed before the paper can be accepted
Comments on the Quality of English LanguageThere is room for improvement. Starting from the third sentence of Abstract, "Although" is not correct. Use "However" or "Nevertheless"
Author Response
Comments and Suggestions for Authors
Although I am still not convinced by the justification of this work nor by the lack of more detailed discussion of the properties of the tested materials, I acknowledge the overall seriousness of the work.
My only but big concern remains the literature citation. While I had previously mentioned the lack of references in the field of DPSCs, the authors have mainly added their own papers. This must be changed before the paper can be accepted
RE: we thank the reviewer for the acknowledgment on the commitment of our study. In introduction section, we have edited the references dealing with DPSCs properties, according to the reviewer's recommendations.
Comments on the Quality of English Language
There is room for improvement. Starting from the third sentence of Abstract, "Although" is not correct. Use "However" or "Nevertheless"
RE: the manuscript has been edited according to the reviewer's suggestions.
Reviewer 2 Report
Comments and Suggestions for Authors
The authors included their improvements in the manuscript, the paper can be considered for publication
Author Response
We thank the reviewer for the acknowledgment.
Round 3
Reviewer 1 Report
Comments and Suggestions for Authors
the new version is acceptable as it is
Comments on the Quality of English Languagefar from being good but acceptable as it is